# Cowpea Immature Pods and Grains Evaluation: An Opportunity for Different Food Sources

**DOI:** 10.3390/plants11162079

**Published:** 2022-08-09

**Authors:** Márcia Carvalho, Valdemar Carnide, Carla Sobreira, Isaura Castro, João Coutinho, Ana Barros, Eduardo Rosa

**Affiliations:** 1Centre for Research and Technology of Agro-Environment and Biological Sciences (CITAB), University of Trás-os-Montes e Alto Douro (UTAD), 5000-801 Vila Real, Portugal; 2Institute for Innovation, Capacity Building and Sustainability of Agri-Food Production (Inov4Agro), University of Trás-os-Montes e Alto Douro (UTAD), 5000-801 Vila Real, Portugal; 3Chemistry Centre (CQ), University of Trás-os-Montes e Alto Douro (UTAD), 5000-801 Vila Real, Portugal

**Keywords:** *Vigna unguiculata* L. Walp., immature pods, immature grains, nutritional quality

## Abstract

Currently, the sustainability of agro-food systems is one of the major challenges for agriculture and the introduction of new pulse-based products can be a good opportunity to face this challenge. Cowpea (*Vigna unguiculata* L. Walp.) is a nutritionally important crop and has the particularity that the aerial section of the plant is entirely edible. The current research determines the nutritional composition of the alternative cowpea food sources immature pods and grains comparatively to dry grains through the evaluation of protein, minerals and different polyphenolic contents, and antioxidant capacity. Ten cowpea genotypes were analyzed during two harvest seasons. Cowpea immature pods and grains revealed high levels of total protein and K, Ca, Zn and Fe contents. In general, most of the genotypes produced cowpea of high nutritional value, with a high variation observed between them. Our results showed the potential of the introduction of new cowpea new products in the market allowing a healthy and variable diet and at the same time a better use of the crop under the scenario of climate change.

## 1. Introduction

Grain legumes have gained significant importance given the need to reduce the consumption of animal food products taking to the increase in the demand for plant based-protein [1]. Cowpea (*Vigna unguiculata* L. Walp) is a warm season grain legume and one of the most important natives from Africa due to its high-quality protein in human consumption [2,3]. Its dry grains have been referred to as a good source of nutrients and phytochemicals, including proteins, minerals (iron and zinc), vitamins (folic acid and vitamin B complex), fibers, resistant starch, unsaturated fatty acids, and phenolic compounds [4,5]. All of the components obtained from the consumption of cowpea bring several protective effects in human health, mainly to chronic diseases, and also to cardiovascular diseases, diabetes, gastrointestinal disorders, obesity and also to several types of cancer [5,6].

Cowpea plays an important role in soil nutrient cycling and green manure through biological nitrogen fixation (BNF) [4,7], with its insertion on the crop rotation system providing an opportunity to increase the productivity of the crops that follow [8,9]. Furthermore, its high adaptability to heat and drought makes it a versatile crop [2] for cultivation, even in the context of global climate change, specifically in Southern Europe, as an increase of temperature and a decrease in rainfall is predicted for the Mediterranean area [10].

This crop is unique in that almost the entire aerial section of the plant is edible, namely the 3 young leaves, immature pods, immature grains and dry grains [11,12]. Traditionally, dry grains are the product that is most consumed worldwide. However, in different parts of the world, the cooked immature pods and grains are also consumed, and are sometimes preferred to the cooked dry grains [13,14]. It is important referred that the best immature pods are the young and slender [13]. The consumption of immature pods and grains as a vegetable is more common in Southeast Asia [2,3], Senegal and some other African countries [14]. In Europe, its cultivation occurs traditionally in some regions of Southern Europe, mainly for production of dry grains, and occasionally in local markets as immature pods [15]. The production of cowpea immature pods and grains could be a good option for more sustainable agriculture, as a reduced water amount is necessary for the formation of these products. Furthermore, its other main advantage is that it will necessitate a much shorter growing season, allowing the crop to escape the climate changes and to be more adapted to several abiotic stresses [3,15,16]. The introduction of novel legume vegetables (immature pods and grains) with higher quality and dietary value into the market will be a valuable option to producers, food industry, vegetable marketing chains, and consumers. Additionally, after all the products have been harvested for food, the rest of the cowpea plant serves as a nutritious fodder for livestock [2,3].

Europe is currently promoting the increase of the consumption of high-protein vegetals using the local agriculture resources and consequently attempting to decrease the deficit of grain legumes production. Southern European countries, namely Portugal, have the perfect weather conditions for the production of cowpea. Its production will promote the decrease of the soil fertilization inputs through the increase of soil fertility and the cowpea insertion in the rotation agriculture will decrease the soil diseases [17]. The use of landraces well adapted to soil and climatic conditions as sources of new products will be beneficial to increase the interest of farmers and consumers. Through there are few scientific publications on this topic, to the best of our knowledge there is no published information on the comparison of the quality and nutritional composition of immature pods and grains and dry grains in Portugal. Therefore, this study intends to compare the protein, minerals (S, B, P, K, Ca, Mg, Zn, Cu, Fe, and Mn) and polyphenolic (total phenolics, flavonoids and *ortho*-diphenols) contents and antioxidant capacity (DPPH and ABTS assays) of 10 cowpea genotypes at three distinct growth stages (immature pods and grains and dry grains) grown in two consecutive years in Portugal. These insights will be useful to promote and instigate the consumption of cowpea immature pods and grains contributing for a healthy and variable diet and at same time to exploit the importance of cowpea production in Southern European countries in the climatic changing scenario decreasing their import in Europe.

## 2. Results and Discussion

The exploitation of new products from high protein crops, including cowpea, for human consumption can be a solution to counteract the healthy food demand scenario and also to contribute to sustainable agriculture. In order to acquire information about cowpea new vegetable products quality, namely immature pods and grains, and also comparing with dry grains, the total phenolic, *ortho*-diphenol and flavonoid contents, antioxidant capacity (ABTS and DPPH), and protein content of ten minerals were determined. Several reports indicate that the cowpea chemical composition may vary considerably according to the accession [6,18]. For this reason, a set of 10 cowpea genotypes previously tested in Portugal [19,20] were used in two consecutive years to assess the best genotype for the production of immature pods and grains and dry grains. The results will be useful in encouraging farmers to produce these new products.

### 2.1. Protein Content

Protein content is an important parameter for the assessment of the nutritional quality of legumes [15,17]. Cowpea, as with other legumes, has a high protein content [4,5], its consumption being a good option to achieve a healthy diet based in high-protein vegetables and raw materials.

Table 1 shows the mean values of protein content obtained from the three types of products of cowpea genotypes in two consecutive years. Significant differences were observed between cowpea products, genotypes and years of experiment (*p* < 0.001; Table 1). In an average of two years, the protein content varied between 233.02 g kg^−1^ (immature pods of Cp5128 genotype) to 313.88 g kg^−1^ (immature grains of BGE038478 genotype). However, the interaction genotype × year did not reveal significant differences.

A slight variation in protein content was observed between the two years of the experiment, with levels in 2017 (208 to 319 g kg^−1^) significantly lower than in 2018 (233 to 343 g kg^−1^) (*p* < 0.001; data not shown). This result could be related with the climatic conditions as the year of 2018 had higher rainfall and lower temperatures (Table 2). Carbas et al. [21] reported the positive influence of these climatic parameters in the protein content in the common bean (*Phaseolus vulgaris* L.).

The results obtained revealed significant differences between the three types of cowpea products (*p* < 0.05), and a high protein content in the new products (immature pods and specially grains), showing their potential to the market and farmers. On average, immature grains had 294.30 g kg^−1^, immature pods 253.08 g kg^−1^ of protein content, while dry grains had an intermediate value of 269.86 g kg^−1^. 

Several studies point the protein contents in dry grains ranging from 220 to 320 g kg^−1^ [4,22]. In this study, the protein content varied from 247.99 g kg^−1^ in the genotype Vg 60 (from Portugal) to 295.21 g kg^−1^ in the genotype BGE038477 (from Spain). Variations between protein contents in these cowpea genotypes have already been obtained in other studies (e.g., [19,20,22,23]). The lowest protein contents were also observed by Dominguez-Perles et al. [23] in dry grains of the genotypes Vg 60 and Vg 63. However, the results previously presented in terms of dry grains’ nutritional composition are very interesting, providing an added value of this legume crop that could be considered well-adapted to the growing conditions in Southern Europe, specifically Portugal [23]. 

Currently, the information about the protein content of cowpea immature pods and grains is still scarce and the comparison between both products and dry grains does not exist. However, some reports from other legume crops present and refer that immature pods and grains of legumes contain lower amounts of protein comparatively to dry grains of the same species [15,16,24]. However, the results obtained in this study revealed that both grain products have higher amounts of protein (294.30 g kg^−1^ immature grains and 269.86 g kg^−1^ dry grains). Contrary to expectations, the immature grains in all genotypes of both years registered higher levels of protein content than dry grains (more than 9%) and immature pods (more than 14%), ranging from 272.48 g kg^−1^ (Cp 5553 genotype) to 313.88 g kg^−1^ (BGE038478 genotype). Despite the fact that immature grains consumption is common in legumes such as pea or faba bean and unusual in cowpea, the results obtained can be considered a good alternative for high-protein vegetables. The immature pods revealed significant differences in comparison to the dry grains (6% less than dry grains), varying from 233.02 g kg^−1^ (Cp5128 genotype) to 272.23 g kg^−1^ (Vg 72 genotype). In studies developed in different conditions but with some similar genotypes, [25] and [15] also observed good scores of protein content using dry weight in immature pods and fresh weight, respectively. Nevertheless, the results obtained in this study allowed us to classify cowpea immature pods as moderately high to rich in protein content when compared to other legumes crops namely green pods of common bean and pea [26,27]. 

The protein content results reveal the huge potential of these new products (immature pods and grains) to be consumed as vegetables. Both products provide a good alternative to the consumers’ diet, allowing for a variable option with a high level of plant-protein. In addition, the introduction of these two new products on the market will be an important resource for cowpea producers, serving as a better use of this crop throughout the different times of harvesting and providing additional income. 

### 2.2. Minerals

For human health and good nutrition, it is necessary a set of minerals to meet metabolic needs, namely calcium (Ca), magnesium (Mg), phosphorus (P) and potassium (K) [13,28]. Normally, these minerals are provided in food intake, and their amount varies with the crop species. Several studies refer to cowpea dry grains as a good source of minerals, especially potassium, calcium, magnesium and phosphorus [6,16,28]. Furthermore, other studies refer to these crops as a source of iron (Fe) and zinc (Zn) [6,29], these mineral elements being useful in the prevention of birth defects during pregnancy [4]. However, [29] refer these elements (Fe and Zn) also have unwanted characteristics because they are responsible for the increase of grain hardness and increased cooking time. The results obtained are presented as mean values for the two years.

In our study, ten mineral elements were quantified in the three cowpea products, including five macrominerals (Ca, K, Mg, P and S) and five microminerals (B, Cu, Fe, Mn and Zn). The results are presented as the average values obtained in both years of field trials (Table 3). Generally, this study revealed significant differences (*p* < 0.001) in the minerals’ contents in the three cowpea products, ten cowpea genotypes and two years of experiments. The highest contents of both types on elements (macro and microminerals) were obtained in 2018, with more pronounced differences between years in immature grains and dry grains (*p* < 0.001, data not shown). The significant differences (*p* < 0.001) obtained between the two years of experiment in the main mineral-elements may be related to climatic conditions, as a higher level of rainfall was recorded in 2018 (Table 2).

The two years’ mean values for each of the five macrominerals (Table 3) ranged (Ca) from 0.60 g kg^−1^ (BGE038477 for calcium in immature grains and Vg60 in dry grains) to 6.10 g kg^−1^ (AUA2 in immature pods); for potassium (K) from 9.70 g kg^−1^ (Cp5128 in immature grains) to 14.90 g kg^−1^ (BGE038477 in immature pods); for magnesium (Mg) from 1.60 g kg^−1^ (Vg 63 in immature grains and AUA2 in dry grains) to 3.40 g kg^−1^ (AUA2 in immature pods); for phosphorus (P) from 3.60 g kg^−1^ (Cp5128 in immature pods) to 5.40 g kg^−1^ (AUA2 in immature grains) and, for sulfur (S) from 0.58 g kg^−1^ (Cp5128 in dry grains) to 1.21 g kg^−1^ (AUA2 in immature pods). In terms of the five microminerals (Table 1), the mean values varied for boron, (B) from 8.90 mg kg^−1^ (Cp5128 in immature grains) to 20.60 mg kg^−1^ (AUA2 in immature pods); for iron (Fe) from 47.90 mg kg^−1^ (Cp5128 in immature pods) to 70.90 mg kg^−1^ (AUA2 in immature grains); for zinc (Zn) from 41.60 mg kg^−1^ (Cp5128 in immature pods and grains) to 64.10 mg kg^−1^ (AUA2 in immature pods); for manganese (Mn) from 12.20 mg kg^−1^ (Fradel in dry grain) to 51.70 mg kg^−1^ (Vg73 in immature pods) and, for cupper (Cu) from 5.30 mg kg^−1^ (Fradel in immature grains) to 10.10 mg kg^−1^ (Vg63 immature pods). 

As has been described in other studies, the K, Ca, Zn and Fe are the four most important cowpea minerals [6,13,30]. This set of minerals plays a significant role in human metabolic health and brings several benefits, including regulating fluid balance and controlling the electrical activity of the heart and other muscles (K), bone formation (Ca), red blood cell formation (Fe) and also fulfills many biochemical metabolism functions (Zn) [13,31]. Data from other cowpea studies have shown potassium (K) to be the most abundant mineral in cowpea dry grains, varying from 1.9 to 28.9 g kg^−1^, while the Ca content varied from 0.38 to 10.62 g kg^−1^, Zn varied from 8.1 to 118 mg kg^−1^, and Fe varied from 6.9 to 218 mg kg^−1^ (reviewed by [6]). The results obtained for the three cowpea products in our study comply with these ranges (average values of the three cowpea products: K = 11.8 g kg^−1^; Ca = 2.1 g kg^−1^; Zn = 47.8 mg kg^−1^; Fe = 58.3 mg kg^−1^) showing the potential of consuming cowpea products as immature pods and grains. Besides these minerals, it should be emphasized that the manganese value in immature pods was much higher (almost four-fold) than the values in immature grains and dry grains. Gerrano et al. [13] also obtained a high level of this mineral in immature pods. Manganese is a micro-element mineral necessary for proper function of the reproductive organs hindering growth [31], consequently the consumption of products rich in this mineral will promote healthy growth. Nevertheless, for the other mineral elements (P, Mg, S, B and Cu), the values obtained in the three cowpea products are in agreement with other studies [13,31,32]. 

Despite the results obtained from all mineral elements being in agreement with other studies, a large genotype variation in the minerals content was observed which was also reported in other studies [13,32]. For the macrominerals, the genotypes BGE038477 and AUA2 revealed to be the richest in the three types of cowpea products, while Cp5128 was the poorest. Genotypes BGE038478 and AUA2 have shown the highest values for microminerals, while Cp5128 and Fradel had lowest. This variation indicates the high biodiversity within this set of genotypes, these being valuable genetic resources to produce new cowpea products (immature grains and pods) interesting to be exploited in vegetables farming systems and consequently in the food industry and market.

Different studies pointed out that immature pods have lower mineral contents compared to grains and leaves [6,33]. However, in our study, the immature pods presented the highest values comparatively to immature grains and dry grains in all the mineral elements contents (approximately 15% more than immature grains and 20% than dry grains) except for phosphorus and iron (Table 3). These results show the potential of these new products (immature pods and grains) as high nutritious vegetables. Their levels of essential minerals (micro- and macrominerals) will allow consumers to maintain a balanced and healthy diet and at the same time to have different options to consume cowpea.

### 2.3. Polyphenolic Content

Phenolic compounds are polyhydroxilated compounds that constitute one of the most abundant groups of plant secondary metabolites [34,35,36], being divided into the subgroups: phenolic acids, flavonoids, tannins and stilbenes [35]. Legumes, including cowpea, are a good source of phenolic compounds. Phenolic compounds are mainly concentrated in the seed coat and are considered natural antioxidants, being responsible for several beneficial effects against chronic diseases [28,34]. Furthermore, these bioactive compounds have an important role in seed pigmentation, growth and reproduction [34,37]. Furthermore, phenolic compounds are involved in the defense from environmental stresses such as drought, salinity, and high and low temperature [37]. Previous studies refer to flavonoids and phenolic acids as the two most common phenolic compounds present in legume seeds, namely in cowpea [28,34].

Total phenolic content, flavonoids and *ortho*-diphenols, in average of the two years the experiment and for the three cowpea aerial organs, are shown in Table 4. The three evaluated parameters are expressed on a dry weight (dw) basis to eliminate variations between samples. In general, significant differences (*p* < 0.05) were observed for the three cowpea products, genotypes and years of experiment. As observed in the other parameters, significant differences were recorded between the two years of experiment (*p* < 0.05), except for the flavonoids content in dry grains (data not shown). As previously mentioned, the different climatic conditions in 2017 and 2018 may explain the annual differences in phenolic compounds contents (Table 2).

The mean values for phenolic content ranged between 1.03 mg GA g^−1^ dw (AUA2 in dry grains) and 20.49 mg GA g^−1^ dw (BGE038478 in immature grains), with an average of 7.93 mg GA g^−1^ dw, while the flavonoid contents varied between 1.10 mg CAT g^−1^ dw (AUA2 in dry grains) and 14.04 mg CAT g^−1^ dw (BGE038478 in immature grains) with an average of 4.73 mg CAT g^−1^ dw and, the *ortho*-diphenols between 2.59 mg GA g^−1^ dw (Vg 72 in dry grains) and 18.51 mg GA g^−1^ dw (BGE038478 in immature pods) with an average of 8.22 mg GA g^−1^ dw (Table 4).

In our data set we observed a high variation between genotypes (*p* < 0.05; Table 4). This result is in accordance with other studies which refer that the phenolic contents vary significantly with the genotypes used in each study [15,21,34,38]. The genotypes BGE038477 and BGE038478 revealed the highest contents in the three phenolic parameters for the three cowpea products, while AUA2 the lowest values (Table 4). Carbas et al. [21] pointed that the dark colored seeds generally reveal high phenolic composition and antioxidant capacity levels. Our results support this affirmation as those genotypes (BGE038477 and BGE038478) have the darker pigmentation (brown seed color; Table 5) and were the richest in the three phenolic parameters, while the genotype AUA2, with the lighter seed pigmentation (Table 5) presented the lower values of phenolic composition.

The phenolic composition obtained for this set of cowpea genotypes and products is equivalent to that obtained in other studies, showing their potential as a good source of bioactive compounds [15,28,34,37,38,39,40]. It is important to mention that the information available about *ortho*-diphenols content in cowpea is scarce, given that it is difficult to establish a suitable comparison. From the three phenolic composition parameters, we verified that higher contents were detected for total phenols and *ortho*-diphenols. This result contradicts previous studies that pointed flavonoids as the phenolic compound in higher concentration in legumes [28,34]. Few studies are available with these new cowpea products, however [15,40] also observed high levels of total phenolic content in immature pods and grains. Our study demonstrated that the immature pods and grains had higher values for phenolic composition in comparison to dry grains, allowing us to affirm the potential of these products for the human diet.

### 2.4. Antioxidant Capacity

Several studies show that the phenolic compounds are directly correlated with the antioxidant properties of cowpea because they act as radical scavengers and reducing agents, establishing a linear correlation with antioxidant capacity [34,37]. Singh et al. [35] refer that legumes rich in phenolic compounds also have a high antioxidant capacity. The daily inclusion of legumes in human diet would be responsible for several beneficial health effects, namely at level of hypoglycaemic, hypolipidaemic, and antihypertensive [35]. Consequently, the evaluation of the antioxidant capacity in cowpea could be an important parameter for the quality assessment of the products. This information is scarce, mainly in immature pods and grains.

Table 6 reports the average results obtained in the years of experiment trials for the measurement of free radical scavenging (ABTS and DPPH) assays in the three products of the ten cowpea genotypes. For the ABTS assay, the mean values ranged from 0.003 mmol Trolox g^−1^ dw (AUA2 in dry grains) to 0.036 mmol Trolox g^−1^ dw (BGE038477 in immature grains), with an average of 0.018 mmol Trolox g^−1^ dw, while the DPPH mean values ranged between 0.216 mmol Trolox g^−1^ dw (BGE038478 in immature grains) to 0.337 mmol Trolox g^−1^ dw (BGE038477 in immature pods) with an average of 0.264 mmol Trolox g^−1^ dw (Table 6). In general, the results obtained are in agreement with the results presented in other studies [15,39,40]. Gan et al. [39] reported that coat pigmented common beans had much higher antioxidant capacity than the most common fruits and vegetables, suggesting that the pigmented beans may be a potential source of natural antioxidants. Considering the set of three cowpea products and two antioxidant capacity methodologies, the genotypes BGE038477 and BGE038478 presented the highest average antioxidant capacity, while the AUA2 and CP5128 genotypes demonstrated the lowest.

In both methodologies, the antioxidant activity of the three cowpea products generally decreased in the second harvest year (data not shown). As has been referred above, this tendency could be explained by the higher average temperatures recorded in 2017 compared to 2018 (Table 2). On the other hand, we also observed a variation of ABTS and DPPH values between genotypes (*p* < 0.001). This variation has also been reported in other cowpea studies, being mainly related with the genetic information of each genotype [15,34]. Until now, few studies have been carried out to evaluate the antioxidant capacity of the three cowpea products (immature pods and grains and dry grain). From this set of samples, the three cowpea products showed similar values in both antioxidant capacity assays (ABTS and DPPH), but the ABTS values of dry grains were low compared to the others. The total antioxidant capacity of the different types of cowpea products obtained in this study can be comparable with other studies [15,34,40] showing that immature pods and grains may exhibit considerable antioxidant properties and suggesting their potential benefit in diets preventing cancer and other chronic diseases [5,34].

## 3. Materials and Methods

### 3.1. Solvents, Chemicals and Standards

Extra pure (>99%) Folin–Ciocalteu’s phenol reagent, acetic acid, potassium hydroxide and sodium hydroxide were purchased from Panreac (Panreac Química S.L.U., Barcelona, Spain). The compounds 2,2-diphenyl-1-picrylhydrazyl (DPPH˙), 2,2-azino-bis (3-ethylbenzothiazoline-6-sulphonic acid) diammonium salt (ABTS), potassium phosphatase were acquired from Sigma-Aldrich (Sigma-Aldrich, Steinheim, Germany). Sodium molybdate (99.5%) was obtained from Chem-Lab (Chem-Lab N.V., Zedelgem, Belgium). Methanol, sodium nitrate, aluminum chloride and sodium carbonate, all extra pure (>99%), were purchased from Merk (Merk, Darmstadt, Germany). All chemicals and reagents used for mineral composition determination were purchased from Sigma-Aldrich (Sigma-Aldrich, Steinheim, Germany) and Merk (Merk, Darmstadt, Germany).

Standards: (+)-Catechin hydrate (>98%) was purchased from Sigma-Aldrich (Sigma-Aldrich, Steinheim, Germany); 3,4,5-trihydroxybenzoic acid (gallic acid) from Panreac (Panreac Química S.L.U., Barcelona, Spain) and; 6-hydroxy-2,5,7,8-tetramethylchroman-2-carboxylic acid (Trolox) from Fluka Chemika (Fluka Chemika, Neu-Ulm, Switzerland).

The ultrapure water used was obtained from a water purification system available in the laboratory (Millipore, Bedford, MA, USA). 

### 3.2. Cowpea Sampling

#### 3.2.1. Plant Material and Experimental Design

Nine cowpea landraces (six from Portugal, two from Spain and one from Greece) and one commercial variety (‘Fradel’, from Portugal) were used in the field experiment (Table 5). This set of genotypes belonging to *Vigna unguiculata* ssp *unguiculata* was selected based on previous studies developed by the group [19,20].

The field experiment was conducted during two consecutive years (spring-summer 2017 and 2018) in an experimental field at the University of Trás-os-Montes and Alto Douro (UTAD), Vila Real, Portugal (41°17′ N, 07°44′ W, 465 m). The genotypes were sown on 16 May 2017 and 9 May 2018 in a randomized complete block with four replicates and a total of 40 plants of each accession per replicate of two row plots with 3 m of length, 0.75 m of row spacing and 4.5 m^2^ of area. Seeds were sown by hand with a seed rate of 11.1 seeds/m^2^. The soil was classified as gleyic fluvisol with a medium texture in both growing seasons and presented in average a total of 1.47 g/kg organic matter, 40 mg/kg of P_2_O_5_, 108 mg/kg of K_2_O_2_ and a pH (KCl) 5.2. The experimental fields were drip irrigated from the middle of June to the beginning of August. During the growing seasons, weeds and pests were hand-controlled.

Throughout the trial (May–August), the average temperature recorded was 20.1 °C in 2017 and 19.9 °C in 2018. The highest average temperature was observed in July and August 2017 and August 2018, while the highest values of rainfall was recorded in May 2017 and June 2018 (Table 2).

#### 3.2.2. Samples Collection and Preparation

Samples of immature pods and grains and dry grains were manually harvested for this study. Sampling was done according to the suitable development stage for each type: the immature pods were harvested when they reached their maximum possible length, however retaining their green color and tenderness (according to described by [15]); the immature grains were harvested when they reached their full-size but did not enter in the maturation stage; the dry grains were harvested from mature pods at the end of the experiment.

A total of 10 pods (approximately 20–40 g of pods) were used per replicate and development stage in each tested genotype. After harvesting, the samples were immediately frozen in liquid nitrogen and maintained at −80 °C until analysis.

For the samples preparation, the immature pods and grains and dry grains collected were lyophilized over five days. Afterwards, the samples were ground to a fine powder and stored at room temperature protected from light until analysis.

### 3.3. Determination of Protein Content

The protein content [41] was estimated using the N content, which was determined by the Kjeldahl method, multiplying it by 6.25.

### 3.4. Mineral Analysis

Phosphorus, potassium, calcium, magnesium, sulfur, boron, iron, copper, zinc and manganese contents wet digestion with HNO_3_ + H_2_O_2_ assisted by microwaves [1] and determined by inductively coupled plasma emission spectrometry (ICP-OES).

### 3.5. Phytochemical Analysis

#### 3.5.1. Polyphenolic Extracts

Three hydro-methanolic extracts were prepared for each sample for the evaluation of phytochemical contents and antioxidant capacity, following the methodologies described by [40]. Each cowpea lyophilized sample (40 mg) was added to 1.5 mL of methanol/distilled water (70:30, *v*/*v*), vortexed, and placed in an orbital shaker at room temperature for 30 min. The mixture was then centrifuged at 10,000 rpm for 15 min at 4 °C and the supernatant was recovered. This procedure was repeated three times, and the final volume was adjusted to 5 mL and stored at 4 °C until spectrophotometric analyses. All the final results were presented in terms of cowpea dry weight (dw).

#### 3.5.2. Determination of Polyphenolic Contents

The polyphenolic contents, namely total phenols, flavonoids and ortho-diphenols, were determined according to spectrophotometric methodologies previously described by [40] for 96-well microplates (Nunc, Roskilde, Denmark) and a Multiscan FC microplate reader (Thermo-Fisher Scientific, Inc., Waltham, MA, USA). For all the analyses, a total of three technical replicates of each sample were assessed. The total phenolic and ortho-diphenol contents were expressed in mg of gallic acid per mg of dry weight (mg GA g^−1^ dw), while the flavonoid content were measured in mg of catechin per mg of dry weight (mg CAT g^−1^ dw).

#### 3.5.3. Evaluation of Antioxidant Capacities

The antioxidant activity was determined using the free radicals DPPH^•^ and ABTS^•+^ spectrophotometric methods adapted to a microscale in accordance with previously described methods by [40]. The measurements were performed using 96-well microplates (Nunc, Roskilde, Denmark) and measured by a Multiscan FC microplate reader (Thermo-Fisher Scientific, Inc., Waltham, MA, USA). A total of three technical replicates of each sample were assessed. The results were expressed as mM Trolox g^−1^ of cowpea tissues dry weight (mM Trolox g^−1^ dw).

### 3.6. Statistical Analysis

All the data were presented as the mean values for the four field replicates (n = 4), each replicate being analyzed in triplicate (n = 3). Differences between means were analysed with one-way ANOVA followed by Tukey’s test at 5% level of significance (*p* < 0.05) using IBM SPSS Statistics version 21.0 software (IBM SPSS, Inc., Chicago, IL, USA). The sample type (immature pods and grains, and dry grains) was defined as the independent variable, while all the parameters assayed were represented as dependent variables, with the loadings corresponding to each one of these parameters being extracted for all of the factors retrieved.

## 4. Conclusions

The introduction of new legume products rich in protein, such as those from cowpea, can be a strategy to promote a healthy diet and at same time to stimulate grain legumes production in Europe, namely in Southern European countries. Increasingly, consumers are looking for diversity and healthy vegetables and an increase in demand for immature pods and grains of legume crops is expected in the future. The present study compared, for the first time, the nutritive value of cowpeas’ immature pods and grains and dry grains.

From the consideration and the analysis of the results obtained, cowpea immature pods and grains proved to be a rich source of protein, mineral and phenolics, also presenting a high antioxidant capacity compared to cowpea dry grains and other legume dry grains. Their consumption will promote a healthy and variable diet. Besides, the introduction of these new products will be also important for the farmers since the immature pods and grains need a shorter growing season compared to dry grains, allowing the crop to escape several abiotic stresses (as drought and high temperatures) imposed by climate change.

Despite the fact that the ten cowpea genotypes tested in this study presented a wide variability for all of the parameters, the immature pods and grains and dry grains showed high quality values for all the parameters studied. These genotypes grown in Southern European countries, namely in Portugal, Spain and Greece, showed their potential to be introduced into the market as high-quality novel legume vegetables. For each parameter evaluated, a genotype was identified with high value, this information important for the breeding and exploitation of new varieties with superior qualities and proprieties. Overall, results suggest that BGE038477 and BGE038478 are the most interesting cowpea genotypes for the production of immature pods and grains of high nutritional value. 

## Figures and Tables

**Table 1 plants-11-02079-t001:** Total protein content of immature pods, immature and dry grains in ten cowpea genotypes (average of two years). Significant differences on genotypes (*p* < 0.05) were evaluated by one and two-way ANOVA, followed by Tukey’s tests.

	Immature Pods	Immature Grains	Dry Grains
Vg 60	249.02	278.42	247.99
Vg 63	260.79	282.65	255.58
Vg 72	272.23	294.07	267.17
Vg 73	247.97	286.33	262.59
Cp 5128	233.02	292.37	281.81
Cp 5553	241.01	272.48	257.23
BGE038477	246.99	310.30	295.21
BGE038478	259.02	313.88	294.66
AUA2	266.31	303.39	259.18
Fradel	254.44	309.08	277.16
Average	253.08	294.30	269.86
%CV	4.69	4.90	6.13
*p* (genotypes)	<0.001	<0.001	<0.001
*p* (year)	<0.001	<0.001	<0.001
*p* (genotype × year)	0.079	0.178	0.404

**Table 2 plants-11-02079-t002:** The average air temperature (°C) and precipitation (mm) per month from May to September in 2017 and 2018 and the 1985–2015 period were recorded at weather station located near the field experiment location.

	Average Temperature			Rainfall		
	2017	2018	1985–2015	2017	2018	1985–2015
May	17.3	15.3	15	108	35	71
June	21.1	18.4	19.2	19	111.8	34
July	21.9	20.6	21.3	22	6.6	14
August	21.9	23.9	22	1.8	0.8	20
September	18.1	21.5	19	0.8	38.11	50
Average	20.06	19.94	19.3	151.6	192.31	189

**Table 3 plants-11-02079-t003:** Two years mean values for the selected macrominerals (P, K, Ca, Mg and S) and microminerals (B, Fe, Zn, Mn and Cu) in immature pods, immature and dry grains of ten cowpea genotypes. Macrominerals expressed in g kg^−1^ and microminerals expressed in mg kg^−1^. Significant differences were evaluated by one-way or two-way ANOVA, followed by Tukey tests for minerals. The absence of a common letter indicates significant differences at *p* < 0.05.

**Macrominerals (g kg^−1^)**					
**Immature Pods**	**P**	**K**	**Ca**	**Mg**	**S**
Vg 60	4.00 ^a,b^	13.30 ^a–d^	4.40 ^a^	3.10 ^a,b^	0.99 ^a,b,c^
Vg 63	4.30 ^b^	13.40 ^b,c,d^	5.10 ^a,b^	3.10 ^a,b^	0.91 ^a^
Vg 72	4.50 ^b^	13.90 ^c,d^	5.20 ^a,b^	3.30 ^b^	1.19 ^b,c^
Vg 73	4.00 ^a,b^	12.30 ^a,b^	4.60 ^a^	3.00 ^a,b^	1.05 ^a,b,c^
Cp 5128	3.60 ^a^	11.70 ^a^	4.00 ^a^	2.90 ^a,b^	0.87 ^a^
Cp 5553	3.90 ^a,b^	13.50 ^a–d^	4.70 ^a,b^	2.70 ^a^	1.00 ^a,b,c^
BGE038477	4.10 ^a,b^	14.90 ^d^	4.00 ^a^	3.10 ^a,b^	0.91 ^a^
BGE038478	4.00 ^a,b^	13.70 ^b,c,d^	4.30 ^a^	3.20 ^a,b^	0.96 ^a,b^
AUA2	4.30 ^b^	13.80 ^b,c,d^	6.10 ^b^	3.40 ^b^	1.21 ^c^
Fradel	3.90 ^a,b^	12.80 ^a,b,c^	4.60 ^a^	3.10 ^a,b^	0.94 ^a,b^
Average	4.07	13.32	4.70	3.10	1.01
%CV	11.54	10.06	21.48	12.58	19.80
*p* (genotypes)	0.003	<0.001	<0.001	0.019	<0.001
*p* (year)	0.001	0.004	0.431	0.014	<0.001
*p* (genotype × year)	0.71	0.436	0.288	0.775	0.867
**Immature grains**	**P**	**K**	**Ca**	**Mg**	**S**
Vg 60	4.40 ^a,b^	10.50 ^a,b^	0.75 ^a^	1.70 ^a,b^	0.69
Vg 63	4.35 ^a^	10.30 ^a,b^	0.76 ^a^	1.60 ^a^	0.61
Vg 72	4.74 ^a–d^	10.90 ^a,b^	0.73 ^a^	1.69 ^a,b^	0.71
Vg 73	4.67 ^a–d^	10.60 ^a,b^	0.64 ^a^	1.71 ^a,b^	0.67
Cp 5128	4.25 ^a^	9.70 ^a^	0.68 ^a^	1.67 ^a,b^	0.59
Cp 5553	4.54 ^a,b,c^	11.00 ^a,b^	0.69 ^a^	1.70 ^a,b^	0.73
BGE038477	5.22 ^b,c,d^	11.50 ^b^	0.60 ^a^	2.00 ^c^	0.72
BGE038478	5.30 ^c,d^	11.56 ^b^	0.72 ^a^	1.92 ^b,c^	0.71
AUA2	5.40 ^d^	11.20 ^a,b^	1.00 ^b^	1.66 ^a^	0.65
Fradel	4.82 ^a–d^	11.44 ^a,b^	0.65 ^a^	1.80 ^a,b,c^	0.74
Average	4.77	10.86	0.72	1.75	0.68
%CV	16.56	13.62	29.16	20.57	16.17
*p* (genotypes)	<0.001	0.022	<0.001	<0.001	0.06
*p* (year)	<0.001	<0.001	<0.001	<0.001	0.001
*p* (genotype × year)	0.38	0.246	0.114	0.574	0.232
**Dry grains**	**P**	**K**	**Ca**	**Mg**	**S**
Vg 60	3.90 ^a^	9.80 ^a^	0.60 ^a^	1.66 ^a,b^	0.73 ^b^
Vg 63	4.27 ^a,b,c^	11.31 ^b,c^	0.78 ^b,c^	1.78 ^b,c^	0.68 ^a,b^
Vg 72	4.28 ^a,b,c^	11.07 ^b,c^	0.81 ^b,c^	1.77 ^a,b,c^	0.67 ^a,b^
Vg 73	4.30 ^b,c^	11.08 ^b,c^	0.74 ^a,b^	1.77 ^a,b,c^	0.73 ^b^
Cp 5128	4.46 ^c^	11.38 ^c^	0.85 ^b,c^	1.84 ^c,d^	0.58 ^a^
Cp 5553	4.25 ^a,b,c^	11.63 ^c^	0.76 ^b^	1.77 ^a,b,c^	0.71 ^b^
BGE038477	4.60 ^c^	11.67 ^c^	0.61 ^a^	1.96 ^d^	0.65 ^a,b^
BGE038478	4.420 ^c^	11.43 ^c^	0.71 ^a,b^	1.96 ^d^	0.67 ^a,b^
AUA2	4.20 ^a,b,c^	10.40 ^a,b^	0.91 ^c^	1.60 ^a^	0.68 ^a,b^
Fradel	3.90 ^a,b^	11.56 ^c^	0.75 ^a,b^	1.83 ^b,c,d^	0.63 ^a,b^
Average	4.26	11.14	0.75	1.79	0.67
%CV	15.72	20.19	26.66	23.46	17.91
*p* (genotypes)	<0.001	<0.001	<0.001	<0.001	0.001
*p* (year)	<0.001	<0.001	<0.001	<0.001	<0.001
*p* (genotype × year)	0.002	<0.001	0.049	0.047	0.077
**Microminerals (mg kg^−1^)**					
**Immature pods**	**B**	**Fe**	**Zn**	**Mn**	**Cu**
Vg 60	16.30 ^a^	60.70 ^c,d^	53.30 ^b,c^	41.00	9.80 ^b,c^
Vg 63	18.00 ^a,b^	61.50 ^c,d^	55.50 ^b,c,d^	46.90	10.10 ^c^
Vg 72	17.30 ^a,b^	65.50 ^d^	58.00 ^c,d^	51.70	9.80 ^b,c^
Vg 73	16.20 ^a^	59.40 ^b,c,d^	54.30 ^b,c^	42.20	8.60 ^b,c^
Cp 5128	16.20 ^a^	47.90 ^a^	41.60 ^a^	36.00	8.80 ^b,c^
Cp 5553	17.70 ^a,b^	52.50 ^a,b,c^	49.40 ^a,b,c^	43.60	8.30 ^b^
BGE038477	18.30 ^a,b^	49.40 ^a,b^	51.30 ^b,c^	48.30	9.90 ^b,c^
BGE038478	18.00 ^a,b^	49.30 ^a,b^	51.10 ^a,b,c^	44.40	9.30 ^b,c^
AUA2	20.60 ^b^	66.70 ^d^	64.10 ^d^	51.30	9.40 ^b,c^
Fradel	19.70 ^a,b^	53.40 ^a,b,c^	46.90 ^a,b^	40.30	6.20 ^a^
Average	17.83	56.64	52.46	44.57	9.03
%CV	10.91	15.75	14.96	25.57	16.05
*p* (genotypes)	0.003	<0.001	<0.001	0.112	<0.001
*p* (year)	<0.001	0.2	0.079	0.093	0.234
*p* (genotype × year)	0.82	0.671	0.984	0.52	0.118
**Immature grains**	**B**	**Fe**	**Zn**	**Mn**	**Cu**
Vg 60	9.76 ^a,b^	56.20 ^a,b^	43.20 ^a^	13.60 ^a^	6.00 ^a,b^
Vg 63	9.91 ^a,b^	53.60 ^a,b^	41.50 ^a^	14.30 ^a,b^	6.20 ^a–d^
Vg 72	9.60 ^a,b^	62.10 ^b,c^	46.20 ^a^	18.10 ^b,c,d^	6.90 ^b–e^
Vg 73	10.00 ^a,b^	59.20 ^a,b^	48.80 ^a,b^	16.00 ^a,b,c^	7.50 ^c,d,e^
Cp 5128	8.90 ^a^	50.10 ^a^	41.60 ^a^	14.60 ^a,b^	6.50 ^a-e^
Cp 5553	10.84 ^a,b,c^	54.60 ^a,b^	43.30 ^a^	15.90 ^a,b,c^	6.10 ^a,b,c^
BGE038477	13.60 ^d^	61.20 ^b,c^	49.00 ^a,b^	21.20 ^d^	7.70 ^e^
BGE038478	13.50 ^d^	60.00 ^b^	47.30 ^a^	20.20 ^c,d^	7.80 ^e^
AUA2	12.70 ^c,d^	70.90 ^c^	55.40 ^b^	21.30 ^d^	7.60 ^d,e^
Fradel	12.00 ^b,c,d^	56.60 ^a,b^	44.90 ^a^	17.80 ^a-d^	5.30 ^a^
Average	11.08	58.44	46.12	17.31	6.75
%CV	20.03	14.35	19.25	22.58	18.37
*p* (genotypes)	<0.001	<0.001	<0.001	<0.001	<0.001
*p* (year)	0.609	<0.001	<0.001	<0.001	0.004
*p* (genotype × year)	0.549	0.898	0.516	0.654	0.034
**Dry grains**	**B**	**Fe**	**Zn**	**Mn**	**Cu**
Vg 60	9.76 ^a,b^	56.20 ^a,b^	43.20	13.60	6.00 ^a,b^
Vg 63	11.06 ^a,b^	60.00 ^a,b^	44.10	13.10	9.10 ^b,c^
Vg 72	10.65 ^a,b^	63.00 ^b^	45.90	13.70	8.96 ^b,c^
Vg 73	11.52 ^a,b,c^	61.90 ^b^	47.00	12.30	9.03 ^b,c^
Cp 5128	10.80 ^a,b^	58.10 ^a,b^	44.70	12.90	8.48 ^a,b,c^
Cp 5553	11.97 ^b,c,d^	58.30 ^a,b^	44.90	13.30	9.23 ^b,c^
BGE038477	12.65 ^c,d^	60.40 ^a,b^	44.50	14.50	9.38 ^c^
BGE038478	12.85 ^d^	63.70 ^b^	47.30	14.00	9.87 ^c^
AUA2	12.67 ^c,d^	61.90 ^b^	45.20	14.30	8.52 ^a,b,c^
Fradel	13.00 ^d^	53.80 ^a^	42.80	12.20	7.30 ^a^
Average	11.78	59.84	44.92	13.36	8.77
%CV	11.03	20.45	18.87	15.19	25.54
*p* (genotypes)	<0.001	0.004	0.184	0.348	<0.001
*p* (year)	<0.001	<0.001	<0.001	0.75	<0.001
*p* (genotype × year)	0.047	0.096	0.025	0.801	0.004

**Table 4 plants-11-02079-t004:** Phenolic content indexes (two year mean values) of three cowpea products (immature pods and grains and dry grains) in the ten genotypes. Total phenolics and *ortho*-diphenols expressed as mg of gallic acid (GA) on dry weight (g); flavonoids expressed as mg of catechin on dry weight (g). Significant differences were evaluated by one-way or two-way ANOVA, followed by Tukey’s tests for each mineral element. Means followed by different letters within the same column are significantly different (*p* < 0.05).

**Immature Pods**	**Total Phenols** **(mg GA g^−^^1^ dw)**	**Flavonoids** **(mg CAT g^−^^1^ dw)**	***Ortho*-Diphenols** **(mg GA g^−^^1^ dw)**
Vg 60	14.29 ^a,b^	5.65 ^b^	15.86 ^b,c^
Vg 63	12.63 ^a,b^	5.72 ^b^	14.55 ^b^
Vg 72	11.99 ^a,b^	5.15 ^b^	15.34 ^b,c^
Vg 73	13.38 ^a,b^	6.25 ^b^	16.74 ^c,d^
Cp 5128	10.13 ^a^	3.27 ^a^	10.39 ^a^
Cp 5553	12.81 ^a,b^	5.10 ^b^	14.53 ^b^
BGE038477	17.78 ^b^	13.40 ^d^	18.51 ^e^
BGE038478	18.26 ^b^	11.43 ^c^	17.64 ^d,e^
AUA2	8.07 ^a^	3.01 ^a^	8.84 ^a^
Fradel	13.35 ^a,b^	6.26 ^b^	14.41 ^b^
Average	13.27	6.52	14.68
%CV	39.11	27.45	23.91
*p* (genotypes)	<0.001	<0.001	<0.001
*p* (year)	<0.001	<0.001	<0.001
*p* (genotype × year)	<0.001	<0.001	<0.001
**Immature grains**			
Vg 60	4.41 ^a,b^	3.52 ^a,b^	3.54 ^b^
Vg 63	4.86 ^a,b^	3.51 ^a,b^	3.80 ^b^
Vg 72	4.98 ^a,b^	3.48 ^a,b^	3.69 ^b^
Vg 73	5.22 ^b^	3.57 ^a,b^	3.70 ^b^
Cp 5128	3.88 ^a^	2.94 ^a,b^	3.40 ^a,b^
Cp 5553	4.94 ^a,b^	4.06 ^b^	3.90 ^b^
BGE038477	19.93 ^d^	13.71 ^c^	15.52 ^d^
BGE038478	20.49 ^d^	14.04 ^c^	15.07 ^d^
AUA2	3.89 ^a^	2.76 ^a^	2.59 ^a^
Fradel	9.58 ^c^	3.97 ^b^	6.18 ^c^
Average	8.22	5.56	6.14
%CV	86.49	80.03	79.47
*p* (genotypes)	<0.001	<0.001	<0.001
*p* (year)	<0.001	<0.001	<0.001
*p* (genotype × year)	<0.001	<0.001	<0.001
**Dry grains**			
Vg 60	1.46 ^a^	1.50 ^b^	3.18 ^a,b^
Vg 63	1.24 ^a^	1.31 ^a,b^	2.73 ^a,b^
Vg 72	1.26 ^a^	1.49 ^b^	2.59 ^a^
Vg 73	1.29 ^a^	1.45 ^b^	2.80 ^a,b^
Cp 5128	1.17 ^a^	1.18 ^a^	3.59 ^b^
Cp 5553	1.27 ^a^	1.48 ^b^	3.15 ^a,b^
BGE038477	6.71 ^b^	5.42 ^d^	7.05 ^c^
BGE038478	6.28 ^b^	5.01 ^c^	6.76 ^c^
AUA2	1.03 ^a^	1.10 ^a^	3.69 ^b^
Fradel	1.31 ^a^	1.19 ^a^	2.99 ^a,b^
Average	2.30	2.11	3.85
%CV	94.78	74.40	55.32
*p* (genotypes)	<0.001	<0.001	<0.001
*p* (year)	<0.001	0.041	<0.001
*p* (genotype × year)	<0.001	<0.001	<0.001

**Table 5 plants-11-02079-t005:** Codes, origin, donor institute, breeding status and seed color information of all the genotypes. * AUA—Agricultural University of Athens, Greece; CRF-INIA—National Plant Genetic Resources Centre-National Institute for Agricultural and Food Technology Research, Spain; INIAV—Instituto Nacional de Investigação Agrária e Veterinária Elvas, Portugal; UTAD—Universidade de Trás-os-Montes e Alto Douro, Portugal.

Code	Origin	Donor Institution *	Breeding Status	Seed Color
Vg 60	Sabugal, Portugal	UTAD	Landrace	Cream with brown eye
Vg 63	Covilhã, Portugal	UTAD	Landrace	Cream with brown eye
Vg 72	Mogadouro, Portugal	UTAD	Landrace	Cream with black eye
Vg 73	Macedo de Cavaleiros, Portugal	UTAD	Landrace	Cream with black eye
Cp 5128	Lardosa, Portugal	INIAV	Landrace	Cream without eye
Cp 5553	Sertã, Portugal	INIAV	Landrace	Cream with brown eye
BGE038477	Malaga, Spain	CRF-INIA	Landrace	Brown with tan brown eye
BGE038478	Malaga, Spain	CRF-INIA	Landrace	Brown with green eye
AUA2	Greece	AUA	Landrace	Cream without eye
Fradel	Portugal	INIAV	Variety	Cream with black eye

**Table 6 plants-11-02079-t006:** In vitro antioxidant activities (DPPH and ABTS), average of two harvest years (mean values, n = 4) of immature pods and grains and dry grains of ten cowpea genotypes. Activities expressed as mmol Trolox g^−1^ dw. Significant differences evaluated by one-way or two-way ANOVA, followed by Tukey tests. Letters indicates significant differences on genotypes (*p* < 0.05).

	ABTS (mmol Trolox g^−1^ dw)			DPPH (mmol Trolox g^−1^ dw)		
	Immature Pods	Immature Grains	Dry Grains	Immature Pods	Immature Grains	Dry Grains
Vg 60	0.0231 ^b^	0.0191 ^b^	0.0057 ^c^	0.2945 ^b,c^	0.2585 ^b,c^	0.2435 ^a^
Vg 63	0.0227 ^b^	0.0213 ^b^	0.0048 ^a,b,c^	0.2882 ^b,c^	0.2528 ^b,c^	0.2417 ^a^
Vg 72	0.0232 ^b^	0.0189 ^b^	0.0042 ^a,b^	0.2943 ^b,c^	0.2489 ^b^	0.2413 ^a^
Vg 73	0.0238 ^b^	0.0194 ^b^	0.0048 ^a,b,c^	0.3061 ^c^	0.2519 ^b^	0.2417 ^a^
Cp 5128	0.0157 ^a^	0.0236 ^b,c^	0.0042 ^a,b,c^	0.2627 ^a^	0.2474 ^b^	0.2408 ^a^
Cp 5553	0.0220 ^b^	0.0271 ^c,d^	0.0054 ^b,c^	0.2940 ^b,c^	0.2525 ^b,c^	0.2431 ^a^
BGE038477	0.0241 ^b^	0.0355 ^e^	0.0215 ^d^	0.3373 ^d^	0.2661 ^c^	0.2581 ^b^
BGE038478	0.0236 ^b^	0.0293 ^d^	0.0211 ^d^	0.3365 ^d^	0.2160 ^a^	0.2561 ^b^
AUA2	0.0200 ^a,b^	0.0102 ^a^	0.0035 ^a^	0.2732 ^a,b^	0.2213 ^a^	0.2410 ^a^
Fradel	0.0231 ^b^	0.0289 ^d^	0.0044 ^a,b,c^	0.3065 ^c^	0.2500 ^b^	0.2431 ^a^
Average	0.0221	0.0233	0.0079	0.2993	0.2466	0.2450
%CV	21.26	60.08	86.07	20.04	11.35	4.89
*p* (genotypes)	<0.001	<0.001	<0.001	<0.001	<0.001	<0.001
*p* (year)	0.021	<0.001	0.002	<0.001	<0.001	<0.001
*p* (genotype × year)	<0.001	<0.001	<0.001	<0.001	<0.001	<0.001

## Data Availability

Not applicable.

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
