# Peer review of "Cowpea Immature Pods and Grains Evaluation: An Opportunity for Different Food Sources"

_plants, 2022, doi:10.3390/plants11162079_

Round 1

Reviewer 1 Report

The manuscript titled "Cowpea immature pods and grains evaluation: an opportunity for different food sources" submitted to MDPI Plants by  Carvalho et al. focuses on the the nutritional composition of the alternative cowpea food sources immature pods and grains comparatively to dry grains through the evaluation of protein, minerals and different polyphenolic contents, and antioxidant capacity. The manuscript is overall well written and can be considered for publication after clearing explained concerns listed below.

Introduction

1. Page 2, line 44-45: Cowpea has the particularity that almost the entire aerial section of plant is edible, 44 namely young leaves, immature pods, immature grains and dry grains. Here reference is required. Check the following references that has been done on the cowpea products:  https://doi.org/10.1080/09064710.2018.1520290,  https://doi.org/10.21273/HORTSCI.50.10.1435.....

2. Line 64-69: Besides, cowpea crop-ping for green pods and immature grains will request a much shorter growing season with fewer inputs than other vegetables, making the crop more sustainable and adapted to several abiotic stresses imposed by climate change (reference is required??). Through there are few scientific publication in different parts of the country, to our best knowledge, there is no published information on the comparison........and dry grains in Portugal.

3. Page 2: Material and methods should come before the results and discussion. Hence, the author needs to move the methodology to page 2.

Page 3: line 103, Fig 1: 'dry grains' is duplicated remove.

Page 4, line 143: use standard citation. The author used the sir names and the numbering for the referencing. .."Machado et al. [19] and Karapanos et al. [9]"...remove the names and be consistence.

Page 5, line 163: remove singh et al and maintain [24]...check throught for the manuscript for the consistency.

Page 10, line 273; line 287: remove names and maintain [15]; [19]; [35]

On the same page, line 295: remove sir name and maintain {30]

page 11: line 310: remove name and maintain [34]

Page 13, line 362: This grouping is in accordance with the results obtained in particular in polyp-phenolic...."remove 'the'

On the same page, line 380: Material methods should move to page 2 before Results and discussion.

In material and methods section, the author of the manuscript needs to include the soil characteristics where the trials were conducted as this can affect the nutritional composition of the cowpea plant products (immature pods, green seeds and dry grains).

Overall, the experiment was well planed and conducted. The presentation of the manuscript was very interesting for the readers specially for the breeders who are interested in breeding for quality.

After correction, the manuscript can be recommended for acceptance.

Reviewer 3 Report

The topic of the article is interesting. However, there are some major revisions needful to get this article accepted.

1. Introduction part is not reflecting the topic properly. The relevance of the keywords of the title, immature pods and grains are completed missing in the introduction part. 

2. I would suggest the author to rewrite the introduction part by including the importance of cowpea as a whole in the changing climatic scenario.

3. Introduction must have the research-gap and research-question. Why such research was done?

4. I would suggest the authors to include the following references in the introduction part:

https://doi.org/10.3390/agronomy11081622

https://doi.org/10.1016/j.indcrop.2019.111710

5. Methodology should come prior to results and discussion.

6. Results are nicely written illustrating proper description of the tables and figures. However, I have found some lacuna in the discussion part. Authors must include the discussion referring the data recorded in the research.

7. Experimental design is not properly mentioned. It is very important part of the research. Authors must illustrate this clearly so that the future readers of this article can understand the design properly.

8. Conclusion part must have the recommendation and take-home message.

Round 2

Reviewer 2 Report

The suggestion on the original draft was to present all the nutritional data in tables showing means and significance of differences. This has not been done for protein content in the revised draft. For ready reference to protein content in different varieties, Fig.1 should be replaced with a table showing the actual protein content as % protein (dry weight basis) which is the standard way to express protein content.

The manuscript should be accepted after this has been done.

Author Response

Thank you for the revision.

The new table with protein data was performed in substitution of figure 1. However the data were presented in g kg-1 due that this unit is more used in the manuscripts with this subject.

Some English revisions was performed.

Reviewer 3 Report

The paper has been improved a lot and can be accepted now.

Author Response

Thank you for the revision.

Some English changes was performed.